

# The pattern and burden of non-communicable diseases in armed conflict-exposed populations in Northeastern Nigeria

Roland I. Stephen[1,2], Jennifer A. Tyndall[3], Jamiu S. Olumoh[4], Malachy I. Okeke[3], Jacob A. Dunga[5], Tonde G. Elijah[6], Dillys M. Bello[7], Oyelola A. Adegboye[8] and Jimmy A. Reyes[9]

[1] Department of Public Health, Modibbo Adama University Teaching Hospital, Yola, Nigeria
[2] School of Doctoral Studies, Unicaf University, Larnaca, Cyprus
[3] Department of Natural and Environmental Sciences, American University of Nigeria, Yola, Nigeria
[4] Department of Mathematics and Statistics, American University of Nigeria, Yola, Adamawa, Nigeria
[5] Department of Medicine, Abubakar Tafawa Balewa Teaching Hospital, Bauchi, Nigeria
[6] Department of Medicine, University of Maiduguri Teaching Hospital, Borno, Nigeria
[7] Department of Nursing, Adamawa State Specialist Hospital, Yola, Adamawa, Nigeria
[8] Menzies School of Health Research, Charles Darwin University, Darwin, Northern Territory, Australia
[9] Department of Nursing and Public Health, University of Northern Iowa, Iowa, United States of America

Corresponding author
Oyelola A. Adegboye,
oyelola.adegboye@menzies.edu.au

## ABSTRACT

**Background**. The risk of non-communicable diseases (NCDs) in conflict and post-conflict settings in Northeastern Nigeria has not been evaluated to date. As this region undergoes recovery, understanding the prevalence of NCDs, such as hypertension, diabetes, depression, and obesity, and the associated behavioral coping mechanisms, is crucial for developing tailored healthcare solutions. Therefore, this study aimed to assess the impact of conflict on the prevalence of NCDs in conflict-exposed areas in Northeastern Nigeria compared with non-conflict regions.

**Methods**. This study was an unmatched cross-sectional study. The participants were selected from inpatients and outpatients at general hospitals in Mubi (conflict-exposed) and Jada (non-conflict), which are local government areas in Adamawa, a state in Northeastern Nigeria. The study was conducted over four months, and data on various health indicators were collected. Multivariable binary logistic regression and complementary log regression were performed to investigate the effects of individual risk factors and regional settings on the prevalence of NCDs.

**Findings**. A sample of 973 individuals from both locations was analyzed. The prevalence of hypertension, diabetes, abdominal obesity, and depression in the entire cohort was 22.92%, 5.04%, 44.19%, and 17.94%, respectively. The rates of hypertension and abdominal obesity in the conflict-exposed Mubi were lower, and the rate of depression was higher than those recorded in Jada. Females showed higher rates of hypertension, obesity, and depression than males. The residents of Mubi had lower odds of having abdominal obesity (adjusted odds ratio (aOR) = 0.18; 95% confidence interval (CI) [0.11–0.28]) but a higher risk of depression (incidence risk ratio (IRR) = 4.78; 95% CI [2.51–9.22]) than those in Jada. However, the participants affected by insurgency

showed higher odds of having both abdominal obesity (aOR = 1.95; 95% CI [1.23–3.08]) and depression (IRR = 1.76; 95% CI [1.08–2.88]) than those who were not affected by the conflict.

**Conclusions**. The findings of this study underscore the urgent need for mental health support in conflict-affected regions and comprehensive healthcare strategies for the aging population. As adjustment of lifestyle factors is crucial for addressing NCDs, effective case management and food security are essential for reducing the risk of NCDs in conflict-exposed populations.

# INTRODUCTION

Northeastern Nigeria is recovering from the impacts of the decade-long insurgency and perennial armed conflict (*Brottem, 2021*; *Garba, 2018*; *Ojiego-Okoro, Umeh & Nwodom, 2024*). During the conflict, healthcare facilities and public infrastructures were destroyed, and private enterprises were decimated (*Tafida et al., 2023*). Between 2008 and 2023, Nigeria has recorded about 8.7 million internally displaced people, forced to live in refugee camps or as internally displaced persons (IDPs) mostly in the northeastern states of Adamawa, Borno and Yobe (*Ekoh et al., 2023*; *Internal Displacement Monitoring Centre, 2024*). The prolonged exposure to these humanitarian crises has inflicted severe psychological distress, economic hardship, and social instability on the affected populations (*Garba, 2018*; *Ojiego-Okoro, Umeh & Nwodom, 2024*). The insurgency has led to the deaths of an estimated 350,000 lives since 2009 (*Internal Displacement Monitoring Centre, 2024*), and the survivors have endured extreme trauma, including the loss of loved ones, forced displacement, lack of food and water, near-death experiences, kidnapping, severe bodily injury, and lack of medicines and social services (*Garba, 2018*; *Ojiego-Okoro, Umeh & Nwodom, 2024*).

NCDs, which include conditions such as diabetes, hypertension, depression, obesity, and cancer, are not primarily caused by an acute infection but have long-term health consequences that require long-term treatment and care (*Allen et al., 2017*; *Avan et al., 2019*; *Carrette, D & Hasumi, 2018*; *Greene-Cramer et al., 2020*; *Hosseinpoor et al., 2012*; *Jawad et al., 2019*; *Narayan, Ali & Koplan, 2010*). Conflict and post-conflict settings, such as those experienced in northeastern Nigeria, increase the risk of developing NCDs (*Carrette, D & Hasumi, 2018*; *Greene-Cramer et al., 2020*; *Jawad et al., 2019*). The stress and trauma of such settings often lead individuals to adopt unhealthy coping mechanisms such as drug abuse, alcohol abuse, smoking, and sedentary behaviors—major risk factors for NCDs (*Ojiego-Okoro, Umeh & Nwodom, 2024*; *Tafida et al., 2023*). The vicious cycle of unhealthy lifestyle habits and socioeconomic disadvantages exacerbates the burden of NCDs in conflict-exposed populations (*Tafida et al., 2023*). Additionally, fragile and

violent situations disrupt the continuity of care for NCDs, weaken health systems and drastically reduce access to social services (*Carrette, D & Hasumi, 2018*).

In Nigeria, the national prevalence of hypertension was estimated at 30.6% (95% confidence interval (CI) [27.3%–34.0%]) (*Adeloye et al., 2021*), with complications such as stroke, chronic kidney disease, heart failure and myocardial infarction affecting about 75% of those diagnosed (*Kassy et al., 2022*). About a quarter of emergency room presentations in urban Nigeria stem from hypertension-related complications (*Adeloye et al., 2021*). Diabetes mellitus (DM) also presents a significant public health challenge, with a pooled national prevalence of 5.77% (95% CI [4.3–7.1]) (*Adeloye et al., 2021*). DM is a chronic metabolic disease associated with factors such as sedentary lifestyle, unhealthy diet, advancing age, low socioeconomic status and depression (*Hill-Briggs et al., 2021*; *Mezuk et al., 2008*; *Ramalan et al., 2023*) and accounts for an overall mortality rate of 30.2 per 100,000 population in Nigeria (*Adeloye et al., 2021*). Other conditions, such as central obesity, affect 39% of the Nigerian population (*Bashir et al., 2022*). Depression, a common mood disorder with a lifetime prevalence of 3.9% in Nigeria (*Gbadamosi et al., 2022*), is particularly debilitating among conflict-exposed populations (*Akinrolie et al., 2022*)

NCDs have a multifactorial origin, with risk factors ranging from psychosocial stress, environmental insults, and genetic predisposition to behavioral and unhealthy cultural practices (*Roberts et al., 2009*; *Sharma et al., 2020*). In Juba, South Sudan, approximately 50% of conflict-exposed individuals met the criteria for depression (*Roberts et al., 2009*). Similar findings have been reported in other conflict-affected areas, such as Ukraine, where there is an increase in blood pressure, consumption of alcohol and cigarettes, and increased incidence of coronary heart disease and endocrine diseases, including DM (*Greene-Cramer et al., 2020*).

To our knowledge, no comprehensive study has been conducted on the pattern of NCDs and their behavioral risk factors during the post-conflict and rebuilding phases of Northeastern Nigeria. Therefore, the objective of this study was to explore the prevalence of NCDs, including hypertension, diabetes, depression, and obesity, and their associated behavioral coping mechanisms in conflict-exposed populations in Northeastern Nigeria. The outcomes of this study may inform healthcare authorities of the patterns and burdens of NCDs in the northeastern region of Nigeria, which may facilitate the allocation of dwindling resources to aid holistic recovery in the face of harsh global economic indices.

## METHODS

### Study areas and population

This study was conducted in two local government areas (counties), Mubi and Jada, located in Adamawa State, northeastern Nigeria. Mubi, which lies in the northern part of Adamawa, was devastated by insurgency that started in 2014 and lasted for approximately seven years. Mubi is a metropolitan town that shares borders with Cameroon and is surrounded by suburban and rural areas. Its inhabitants include farmers, traders, civil servants, and herders.

Jada is located in the southern part of Adamawa State. It was largely isolated from the terrorist attacks and guerrilla warfare that plagued northeastern Nigeria. It is a suburban

town surrounded by villages, and its inhabitants are largely farmers, herders, traders, and a small population of civil servants.

## Study design, sample size, sampling

We conducted an unmatched cross-sectional study of inpatients and outpatients who visited the general outpatient clinics of general hospitals in Jada and Mubi over four months, from December 2022 to March 2023. These facilities do not have specialty clinics. To sample patients, we used a combination of simple random sampling and convenience sampling. Participants were selected as they presented to the clinics and were randomly chosen from those admitted to the wards. We selected patients aged between 18 and 100 years and excluded pregnant women, women in puerperium, and individuals with impaired cognition. In addition, we excluded patients judged as frail or non-ambulatory. Data were collected using questionnaires administered by trained data collectors.

We calculated the sample size for this study using Fischer's formula $n = \frac{Z^2 p(p-1)}{d^2}$ at a 95% confidence level based on the 38.4% prevalence ($p$) of hypertension reported in a similar study conducted in Ukraine (*Greene-Cramer et al., 2020*), $d$ is the absolute desired precision at 5%. The calculation was performed at a 0.05 degree of accuracy, and the results indicated a sample size of 363. Although the required sample size for hypertension, with an estimated prevalence of 30.6% (*Adeloye et al., 2021*), was 327 participants, the sample size for diabetes mellitus, with a lower prevalence of 5.8% (*Adeloye et al., 2021*), was calculated to be 85 participants. For depression, with a prevalence of 3.9% (*Gbadamosi et al., 2022*), a sample size of about 58 participants was required, and for obesity, which has a prevalence of 39% (*Bashir et al., 2022*), approximately 367 participants were necessary. However, to increase the power of the study and ensure sufficient representation for all non-communicable diseases (NCDs) being investigated, our final sample size was set at 463 participants, with an equal number of participants recruited from both Jada and Mubi.

Ethical approval for this study was obtained from the National Health Research Ethics Committee, Federal Ministry of Health, Abuja, Nigeria (approval number: NHREC/01/01/2007-03/05/2023). All the participants provided written informed consent prior to participation in the study.

## Study procedure

A structured questionnaire, divided into sociodemographic, NCD risk profile, and anthropometric sections, was administered by trained data collectors using a secure software platform. Participants presenting to the clinics in both study locations were randomly selected as they visited the hospital for healthcare at the general outpatient clinic. They were approached and informed about the research. Their informed consents were obtained. The questionnaires were then administered by the trained data enumerators in the consulting room under confidentiality. The data enumerators were trained on how to interpret and administer the content of the questionnaire in the local dialect and accurately measure the hip and waist circumferences. Blood pressure, blood sugar level, and waist and hip circumferences were measured using the above-mentioned standard procedures. Data on sociodemographic characteristics, including sex, marital status, occupation, and

educational history, were obtained. In addition, data on personal and family histories of NCDs were obtained.

## Study covariates and measurements

**Hypertension** was defined using the World Health Organization (WHO) cutoff, which is a blood pressure of ≥140/90 mmHg obtained in two separate measurements performed using standardized and validated procedures 30 min apart after resting for at least 5 min or self-reported use of anti-hypertension medications (*Gabb et al., 2016*; *Rodgers et al., 2017*; *World Health Organization, 2021*).

**Diabetes** was defined using the American Diabetes Association criteria, which is new-onset diabetes (no history of diabetes) with a fasting plasma glucose level [FPG] of ≥7.0 mmol/L or a random blood glucose level of ≥11.1 mmol/L in addition to the presence of osmotic symptoms (*American Diabetes Association, 2017*; *Zhang et al., 2022*). Prediabetes was defined as FPG of 5.6–6.9 mmol/l. Capillary blood samples were used for the measurement of blood glucose levels.

**Waist circumference** was measured using a flexible tape wrapped around the body at the midpoint between the ribs and the iliac crest after exhalation (*Okeahialam et al., 2012*). Central obesity was defined using the adult waist-hip ratio (WHR) (male WHR, ≥0.90; female WHR, ≥0.85) (*Bashir et al., 2022*).

**Hip circumference** was measured using a flexible tape wrapped around the body at the point where the buttocks extend the most when viewed from the side (*Okeahialam et al., 2012*).

**Fruit and vegetable intake** was determined based on the number of servings typically consumed daily. Five or more servings (at least 400 g) are considered sufficient and fewer than five servings were considered insufficient (*World Health Organization, 2013*).

**Physical inactivity** was defined as failure to meet the WHO recommendations regarding physical activity for health, which is defined as engaging in at least 150 min of moderate-intensity activity per week or 75 min of vigorous-intensity activity per week, by engaging in a combination of walking and moderate- or vigorous-intensity activities (*Bull et al., 2020*).

**Salt consumption** estimates were obtained using the WHO steps instrument (*World Health Organization, 2008*).

**Depression** was assessed using the Patient Health Questionnaire (PHQ-2), which is a depression screening tool. The scores of the PHQ-2 range from 0–6 points, and a score ≥3 suggests the presence of depression. We classified the participants as having depression or no depression based on a cut-off of 3 points (*Maurer, 2012*).

**Socioeconomic status** was categorized using the revised Kuppuswamy socioeconomic status class classification (*Wani, 2019*), which considers three key factors: the education level of the head of the family, occupation and monthly income. Each parameter Is assigned a score, and the total score (ranging from 1 to 29) determines the family's socioeconomic class. The classes are categorized as follows: upper: 26–29, upper middle (16–25), middle (11–15), lower middle (5–10), and lower class (<5).

## Statistical analysis

We performed multivariable binary logistic regression (MBLR) to model the outcomes of each dependent variable (non-communicable diseases, abdominal obesity, and hypertension) as a function of the independent variables (location and impact of insurgency attacks), while adjusting for the covariates (patient background characteristics such as age, sex, marital status, educational level, occupation type, socioeconomic status, health insurance, smoking status, snuffing status, alcohol consumption, daily salt intake, regular physical exercise, family history of hypertension, and family history of diabetes).

Given the low number of cases of depression and diabetes in this study, a multivariate complementary log–log (MClog–log) regression model was used to model the outcomes of patients with depression and diabetes, while adjusting for other covariates. MClog–log is an acceptable alternative to binary logistic regression when the probability of an event is relatively small or large; that is when the outcome is extremely skewed. The regression coefficients from both the MBLR and MClog–log models are presented as adjusted odds ratios (aOR) and adjusted incidence rate ratios (IRR), respectively, along with their 95% confidence intervals. Statistical significance was set at $p < 0.05$. The analyses were performed using Stata Version 13.0.

## RESULTS

A total of 973 participants were included in this study. The demographic characteristics and health indicators of the participants from conflict-exposed Mubi compared with those of the participants from Jada (comparison) are presented in Table 1. The proportion of the study cohort from Mubi (conflict-exposed) (52.93%) was slightly higher than that from Jada (comparison) (47.07%). The average age of the participants was 41.04 years. The average age of the participants from Jada (comparison) was slightly older than that of those from Mubi ($p = 0.020$). The sex distribution of the study population was balanced overall, with no statistically significant difference between Mubi (conflict-exposed) and Jada (comparison) ($p = 0.510$). There were no disparities in marital status between Mubi and Jada; however, the percentage of divorced individuals in Jada (comparison) (3.06%) was higher than that in Mubi. There were significant disparities in educational attainment and occupation between participants from Mubi and those from Jada ($p < 0.001$). Socioeconomic status varied significantly between the two locations, with a higher proportion of upper-income individuals in Jada (comparison) and more low-income individuals in Mubi. Abdominal obesity was more prevalent in Jada (comparison) (58.95%) than in Mubi (31.07%). Although there was no significant difference in blood pressure distribution between the participants from Mubi and those from Jada, there was a significant difference in the prevalence of depression in both regions ($p < 0.001$), with the prevalence in Mubi (conflict-exposed) being higher than that in Jada. There was no difference in glycemic status between the two regions ($p = 0.502$). However, there was a significant difference in healthcare insurance coverage between the two regions ($p < 0.001$), the percentage of individuals without insurance in Mubi (conflict-exposed) is higher than that in Jada, suggesting potential barriers to healthcare access in conflict-exposed regions. There are more civil servants in Mubi than there are Jada (27.82% vs 9.83%) (Table 1).

**Table 1  Characteristics of participants included in the study.**

| Variable | N | % | Location Mubi (conflict-exposed) | Jada (comparison) | P value* |
|---|---|---|---|---|---|
| Overall | 973 | 100 | 515 (52.93) | 458 (47.07) | |
| Age (SD) | | 41.04 (15.74) | 42.15 (15.57) | 39.79 (15.84) | 0.020 |
| **Age group** | | | | | |
| 18–37 | 451 | 46.35 | 228 44.27) | 223 (48.69) | |
| 38–57 | 364 | 37.41 | 205 (39.81) | 159 (34.72) | 0.028* |
| 58–77 | 139 | 14.29 | 67 (13.01) | 72( 15.72) | |
| 78–98 | 19 | 1.95 | 15 (2.91) | 4 (0.87) | |
| **Sex** | | | | | 0.510 |
| Female | 516 | 53.03 | 268 (52.04) | 248 (54.15) | |
| Male | 457 | 46.97 | 247 (47.96) | 210 (45.85) | |
| **Marital status** | | | | | 0.091 |
| Divorce/Separated | 21 | 2.16 | 7 (1.36) | 14 (3.06) | |
| Married | 708 | 72.76 | 389 (75.53) | 319 (69.65) | |
| Single | 185 | 19.01 | 91 (17.67) | 94 (20.52) | |
| Widowed | 59 | 6.06 | 28 (5.44) | 31 (6.77) | |
| **Highest level of education** | | | | | <0.001* |
| No formal education | 343 | 35.25 | 110 (21.36) | 233 (50.87) | |
| Primary | 105 | 10.79 | 76 (14.76) | 29 (6.33) | |
| Secondary | 238 | 24.46 | 137 (26.6) | 101 (22.05) | |
| Tertiary | 287 | 29.5 | 192 (37.28) | 95 (20.74) | |
| **Occupation** | | | | | <0.001* |
| Artisan | 53 | 5.45 | 38 (7.39) | 15 (3.28) | |
| Civil servant | 188 | 19.34 | 143 (27.82) | 45 (9.83) | |
| Farmer | 188 | 19.34 | 90 (17.51) | 98 (21.40) | |
| Full-time housewife | 89 | 9.16 | 33 (6.42) | 56 (12.23) | |
| Herder | 21 | 2.16 | 1 (0.19) | 20 (4.37) | |
| Students | 60 | 6.17 | 35 (6.81) | 25 (5.46) | |
| Trader/Business | 270 | 27.78 | 135 (26.26) | 135 (29.48) | |
| Others | 103 | 10.61 | 39 (7.58) | 64 (13.98) | |
| **Socioeconomic status** | | | | | <0.001* |
| Low income | 735 | 75.54 | 346 (67.18) | 389 (84.93) | |
| Middle income | 223 | 22.92 | 160 (31.07) | 63 (13.76) | |
| Upper income | 15 | 1.54 | 9 (1.75) | 6 (1.31) | |
| **Abdominal obesity** | | | | | <0.001* |
| Normal | 543 | 55.81 | 355 (68.93) | 188 (41.05) | |
| Obesity | 430 | 44.19 | 160 (31.07) | 270 (58.95) | |
| **Blood pressure** | | | | | 0.092 |
| Hypertensive | 223 | 22.92 | 107 (20.78) | 116 (25.33) | |
| Normal | 750 | 77.08 | 408 (79.22) | 342 (74.67) | |

**Table 1** (*continued*)

| Variable | N | % | Location Mubi (conflict-exposed) | Jada (comparison) | P value* |
|---|---|---|---|---|---|
| **Depression** | | | | | <0.001* |
| Negative | 796 | 82.06 | 360 (70.31) | 436 (95.20) | |
| Positive | 174 | 17.94 | 152 (29.69) | 22 (4.80) | |
| **Glycemic status** | | | | | 0.502 |
| Diabetic blood sugar level | 49 | 5.04 | 28 (5.44) | 21 (4.59) | |
| Impaired fasting glycemic | 38 | 3.91 | 17 (3.30) | 21 (4.59) | |
| Normal blood sugar level | 886 | 91.06 | 470 (91.26) | 416 (90.83) | |
| **Healthcare insurance** | | | | | <0.001* |
| No | 915 | 94.04 | 469 (91.07) | 446 (97.38) | |
| Yes | 58 | 5.96 | 46 (8.93) | 12 (2.62) | |

**Notes.**
*Calculated using the chi-square test. *p* value <0.05 is significant.

Table 2 shows the prevalence of chronic conditions in the study population. The prevalence of hypertension, diabetes, abdominal obesity, and depression in the study population was 22.92%, 5.04%, 44.19%, 17.94%, respectively. The participants in conflict-exposed Mubi exhibited a lower prevalence of hypertension (20.78%) and abdominal obesity (31.07%) and a higher prevalence of depression (29.69%) than their counterparts in Jada (comparison) (25.33% and 58.95%, respectively). Females showed higher rates of hypertension (24.81%), abdominal obesity (52.71%), and depression (18.64%) than males. However, males showed a higher prevalence of diabetes (6.13%) than the females. The prevalence of various conditions was notably higher among divorced and widowed individuals than among the married participants. Lifestyle factors such as smoking, sedentary behavior, high salt intake, alcohol consumption, and non-compliance with the WHO dietary recommendations also indicate varying prevalence rates for different chronic conditions. Family history also plays a role, indicating potential genetic predispositions.

The results of the multivariable logistic regression analysis of the effects of explanatory variables on NCDs, adjusted for risk factors, are shown in Table 3. The participants from the conflict-exposed Mubi exhibited significantly lower odds of abdominal obesity than their counterparts in Jada (comparison) (adjusted odds ratio [aOR] = 0.18; 95% confidence interval CI [0.11–0.28]). However, the participants from Mubi showed a higher risk of depression than those from Jada (adjusted incidence risk ratio [IRR] = 4.78; 95% CI [2.51–9.12]). Conversely, the participants in the study showed significantly higher odds of having abdominal obesity (aOR = 1.94; 95% CI [1.51–3.08]), along with an elevated risk of depression (IRR = 1.76; 95% CI [1.08–2.88]).

Age was found to be a significant predictor across all the four non-communicable diseases (NCDs) studied. Participants aged 38–57 had more than twice the odds of having abdominal obesity (aOR = 2.08, 95% CI [1.46–2.96]), nearly double the odds of developing hypertension (aOR = 1.85, 95% CI [1.21–2.83]), and more than twice the risk of developing depression (IRR = 2.46, 95% CI [1.60–3.79]) compared to those aged 18–37. The oldest age group (78–100 years) exhibited significantly higher odds of hypertension (aOR = 4.39, 95% CI [1.64–11.77]) and an increased risk of diabetes (IRR = 4.43, 95% CI [0.43–48.25]).

**Table 2** Prevalence (n [%]) of chronic conditions categorized according to sociodemographic characteristics.

| Variables | Hypertension | Diabetes | Abdominal obesity | Depression |
|---|---|---|---|---|
| **Overall** | 223 (22.92) | 49 (5.04) | 430 (44.19) | 174 (17.94) |
| **Location** | | | | |
| Mubi | 107 (20.78) | 28 (5.44) | 160 (31.07) | 152 (29.69) |
| Jada (control) | 116 (25.33) | 21 (4.59) | 270 (58.95) | 22 (4.80) |
| **Sex** | | | | |
| Female | 128 (24.81) | 21 (4.07) | 272 (52.71) | 96 (18.64) |
| Male | 95 (20.79) | 28 (6.13) | 158 (34.57) | 78 (17.14) |
| **Marital status** | | | | |
| Divorced | 6 (30.00) | 2 (10.00) | 14 (70.00) | 2 (10.00) |
| Married | 176 (24.86) | 43 (6.07) | 316 (44.63) | 138 (19.55) |
| Single | 13 (7.03) | 1 (0.54) | 62( 33.51) | 22 (11.96) |
| Widowed | 28 (47.46) | 3 (5.08) | 38 (64.41) | 12 (20.34) |
| **Smoking** | | | | |
| No | 218 (23.09) | 47 (4.98) | 421 (44.60) | 168 (17.85) |
| Yes | 5 (17.24) | 2 (6.90) | 9 (31.03) | 6 (20.69) |
| **Sedentary lifestyle (physical exercise in line with WHO recommendations)** | | | | |
| No | 134 (25.77) | 25 (4.81) | 240 (46.15) | 81 (15.61) |
| Yes | 89 (19.65) | 24 (5.30) | 190 (41.94) | 93 (20.62) |
| **Daily salt intake** | | | | |
| <5 g/day | 111 (22.33) | 24 (4.83) | 229 (46.08) | 65 (13.10) |
| >5 g/day | 112 (23.53) | 25 (5.25) | 201 (42.23) | 109 (23.00) |
| **Alcohol consumption** | | | | |
| No | 202 (22.30) | 47 (5.19) | 397 (43.82) | 168 (18.60) |
| Yes | 21 (31.34) | 2 (2.99) | 33 (49.25) | 6 (8.96) |
| **Intake of fruits and vegetables (taking up to five fruit servings daily as recommended by the WHO)** | | | | |
| No | 218 (22.88) | 49 (5.14) | 424 (44.49) | 174 (18.32) |
| Yes | 5 (25.00) | 0 | 6 (30.00) | 0 |
| **Family history of hypertension** | | | | |
| No | 165 (19.88) | 31 (3.73) | 363 (43.73) | 142 (17.17) |
| Yes | 58 (40.56) | 18 (12.59) | 67 (46.85) | 32 (22.38) |
| **Family history of diabetes** | | | | |
| No | 200 (22.35) | 36 (4.02) | 395 (44.13) | 153 (17.15) |
| Yes | 23 (29.49) | 13 (16.67) | 35 (44.87) | 21 (26.92) |

Additionally, sex-specific differences were observed, with males having lower odds of abdominal obesity than females (aOR = 0.41, 95% CI [0.29–0.59]). Educational level was a significant predictor of hypertension, where participants with tertiary education had lower odds of hypertension (aOR = 0.43, 95% CI [0.23–0.78]) compared to those with no formal education. Socioeconomic status also played a role in hypertension, with participants in the upper income level having more than four times the odds of developing hypertension compared to those in lower income levels (aOR = 4.36, 95% CI [1.13–16.80]).

**Table 3 Multivariable logistic regression analysis of the effects of explanatory variables on NCDs.**

| Variables | Abdominal obesity | | Hypertension | | Depression | | Diabetes | |
|---|---|---|---|---|---|---|---|---|
| | aOR | 95% CI | aOR | 95% CI | IRR | 95% CI | IRR | 95% CI |
| *Explanatory variables* | | | | | | | | |
| **Location** | | | | | | | | |
| Jada (Reference) | 1 | | 1 | | 1 | | 1 | |
| Mubi (conflict-exposed) | 0.18*** | [0.11, 0.28] | 0.82 | [0.50, 1.34] | 4.78*** | [2.51, 9.12] | 2.00 | [0.82, 4.86] |
| **Affected by insurgency** | | | | | | | | |
| No | 1 | | 1 | | 1 | | 1 | |
| Yes | 1.94** | [1.23, 3.08] | 1.03 | [0.63, 1.68] | 1.76* | [1.08, 2.88] | 0.33 | [0.13, 0.80] |
| *Control variables* | | | | | | | | |
| **Age** | | | | | | | | |
| 18–37 | 1 | | 1 | | | | | |
| 38–57 | 2.08*** | [1.46, 2.96] | 1.85** | [1.21, 2.83] | 2.46*** | [1.60, 3.79] | 5.71** | [2.01, 16.23] |
| 58–77 | 1.87* | [1.12, 3.14] | 2.63*** | [1.52, 4.55] | 1.70 | [0.96, 3.02] | 11.92*** | [3.27, 43.42] |
| 78–98 | 1.04 | [0.36, 2.98] | 4.39** | [1.64, 11.77] | 0.38 | [0.05, 2.68] | 4.56 | [0.43, 48.25] |
| **Sex** | | | | | | | | |
| Female | 1 | | 1 | | 1 | | 1 | |
| Male | 0.41*** | [0.29, 0.59] | 1.06 | [0.71, 1.58] | 0.80 | [0.56, 1.14] | 1.80 | [0.88, 3.66] |
| **Marital status** | | | | | | | | |
| Divorced | 1 | | 1 | | 1 | | 1 | |
| Married | 0.52 | [0.21, 1.30] | 0.74 | [0.27, 2.02] | 0.94 | [0.26, 3.37] | 0.50 | [0.09, 2.74] |
| Single | 0.60 | [0.23, 1.58] | 0.40 | [0.12, 1.29] | 1.02 | [0.26, 4.10] | 0.14 | [0.01, 2.48] |
| Widowed | 0.59 | [0.20, 1.76] | 1.32 | [0.42, 4.19] | 0.92 | [0.23, 3.67] | 0.24 | [0.03, 1.71] |
| **Education level** | | | | | | | | |
| No formal education | 1 | | 1 | | 1 | | 1 | |
| Primary school | 1.07 | [0.65, 1.76] | 0.92 | [0.53, 1.60] | 0.93 | [0.56, 1.54] | 0.90 | [0.35, 2.32] |
| Secondary | 0.75 | [0.50, 1.14] | 0.63 | [0.39, 1.00] | 0.90 | [0.56, 1.45] | 1.75 | [0.75, 4.06] |
| Tertiary | 0.65 | [0.41, 1.03] | 0.43** | [0.23, 0.78] | 1.23 | [0.75, 2.04] | 1.20 | [0.46, 3.17] |
| **Occupation** | | | | | | | | |
| Artisan | 1 | | 1 | | 1 | | 1 | |
| Civil Servant | 1.42 | [0.63, 3.20] | 1.72 | [0.66, 4.48] | 0.94 | [0.41, 2.14] | 2.25 | [0.79, 6.47] |
| Farmer | 0.93 | [0.43, 2.02] | 1.39 | [0.57, 3.39] | 1.60 | [0.75, 3.38] | 1.06 | [0.37, 3.00] |
| Full time housewife | 0.87 | [0.37, 2.04] | 1.79 | [0.68, 4.71] | 1.40 | [0.55, 3.55] | 3.22 | [0.93, 11.17] |
| Herder | 0.72 | [0.23, 2.31] | 1.45 | [0.35, 5.93] | 1 | | 1.61 | [0.17, 15.73] |
| Other | 0.89 | [0.38, 2.07] | 1.24 | [0.47, 3.27] | 1.88 | [0.80, 4.41] | 1.88 | [0.61, 5.81] |
| Student | 0.51 | [0.19, 1.40] | 0.5 | [0.08, 2.96] | 0.71 | [0.23, 2.22] | 1 | |
| Trader/Business | 1.25 | [0.59, 2.63] | 1.26 | [0.52, 3.07] | 1.64 | [0.78, 3.42] | 1 | |
| **Socioeconomic status** | | | | | | | | |
| Low income | 1 | | 1 | | 1 | | 1 | |
| Middle income | 0.82 | [0.54, 1.23] | 0.88 | [0.54, 1.43] | 0.95 | [0.62, 1.47] | 0.54 | [0.26, 1.13] |
| Upper income | 2.01 | [0.65, 6.26] | 4.36* | [1.13, 16.80] | 2.41 | [0.92, 6.36] | 2.73 | [0.34, 22.19] |

**Table 3** (*continued*)

| Variables | Abdominal obesity | | Hypertension | | Depression | | Diabetes | |
|---|---|---|---|---|---|---|---|---|
| | aOR | 95% CI | aOR | 95% CI | IRR | 95% CI | IRR | 95% CI |
| **Health insurance** | | | | | | | | |
| No | 1 | | 1 | | 1 | | 1 | |
| Yes | 0.88 | [0.45, 1.70] | 0.96 | [0.47, 1.97] | 1.12 | [0.58, 2.17] | 0.72 | [0.24, 2.14] |
| **Smoking status** | | | | | | | | |
| No | 1 | | 1 | | 1 | | 1 | |
| Yes | 0.51 | [0.22, 1.18] | 0.51 | [0.14, 1.85] | 1.67 | [0.76, 3.71] | 0.82 | [0.12, 5.66] |
| **Snuff usage** | | | | | | | | |
| No | 1 | | 1 | | 1 | | 1 | |
| Yes | 1.61 | [0.65, 3.98] | 0.19 | [0.02, 1.66] | 3.13[*] | [1.31, 7.49] | 1.46 | [0.31, 6.79] |
| **Alcohol consumption** | | | | | | | | |
| No | 1 | | 1 | | 1 | | 1 | |
| Yes | 1.12 | [0.63, 1.99] | 1.66 | [0.89, 3.12] | 0.50 | [0.23, 1.09] | 0.26 | [0.02, 2.84] |
| **Daily fruit intake** | | | | | | | | |
| No | 1 | | 1 | | 1 | | 1 | |
| Yes | 0.75 | [0.24, 2.31] | 1.26 | [0.38, 4.13] | 1 | | 1 | |
| **Daily salt intake** | | | | | | | | |
| <5g/day | 1 | | 1 | | 1 | | 1 | |
| >5g/day | 1.05 | [0.79, 1.40] | 1.16 | [0.83, 1.63] | 1.40 | [0.99, 1.97] | 1.25 | [0.68, 2.30] |
| **Regular physical exercise** | | | | | | | | |
| No | 1 | | 1 | | 1 | | 1 | |
| Yes | 1.19 | [0.88, 1.62] | 0.90 | [0.62, 1.32] | 1.16 | [0.82, 1.63] | 1.65 | [0.81, 3.33] |
| **Family history of hypertension** | | | | | | | | |
| No | 1 | | 1 | | 1 | | 1 | |
| Yes | 1.07 | [0.70, 1.63] | 3.12[***] | [2.00, 4.88] | 1.3 | [0.81, 2.09] | 2.96[**] | [1.39, 6.31] |
| **Family history of diabetes** | | | | | | | | |
| No | 1 | | 1 | | 1 | | 1 | |
| Yes | 1.04 | [0.60, 1.79] | 0.95 | [0.52, 1.73] | 1.04 | [0.59, 1.84] | 4.31[***] | [1.92, 9.67] |
| **Model diagnostics** | | | | | | | | |
| Number of observations | 972 | | 972 | | 929 | | 807 | |
| Model *vs.* empty model (Wald or LR) | F(30, 972) = 143.74; $p < 0.001$ | | F(30, 972) = 110.77; $p < 0.001$ | | F(28, 929) = 158.06; $p < 0.001$ | | F(27, 807) = 102.47; $p < 0.001$ | |
| Goodness-of-fit test | Hosmer-Lemeshow: F(8, 972) = 10.54; $p = 0.229$ | | Hosmer-Lemeshow: F(8, 972) = 16.02; $p = 0.042$ | | | | | |
| Area under ROC curve | 0.739 | | 0.752 | | | | | |

**Notes.**
[*]$p < 0.05$.
[**]$p < 0.01$.
[***]$p < 0.001$.

Family histories of hypertension and diabetes were significant predictors of diabetes and hypertension, respectively. Family history was associated with higher odds of developing hypertension (aOR = 3.12, 95% CI [1.82–4.88]) and an increased risk of diabetes (IRR = 2.96, 95% CI [1.39–6.31]). Additionally, participants with a family history of diabetes had a higher risk of developing diabetes (IRR = 4.31, 95% CI [1.92–9.67]).

In general, the model diagnostics indicate that the four regression models significantly improve prediction compared to an empty model, with all $p$-values <0.001. The goodness-of-fit test shows that the abdominal obesity model fits the data well ($p = 0.229$), while the Hypertension model reveals some issues with the fit ($p = 0.042$), suggesting that the predicted probabilities do not perfectly match the observed outcomes. The area under the ROC curve (AUC) for abdominal obesity (0.739) and hypertension (0.752) suggests that both models have acceptable discriminatory power, effectively distinguishing between individuals with and without the outcomes. Overall, while the models generally perform well, the hypertension fit could be improved.

## DISCUSSION

This research conducted in the conflict-exposed Northeastern Nigeria outlines the distinct differences in several key indicators between two locales in the region: Mubi, a conflict-affected area, and Jada, a territory that was largely shielded from the insurgency that plagued the region. One of the most prominent differences in the health indicators for NCDs was the five-fold higher risk of depression among the residents of the conflict-exposed Mubi than among those in Jada.

About five years post-conflict exposure in Plateau State, northcentral Nigeria, the prevalence of depression amongst affected persons was found to be 56.3% (*Davou et al., 2018*). This is in consonant with the findings of this study. While the conflict-exposed population of Mubi recorded a 30% prevalence of depression in the post-conflict phase, the national prevalence of depression in Nigeria is 3.8% (*Gbadamosi et al., 2022*). Post-conflict depression represents the most consistent mental health complication of conflict (*Anbesaw et al., 2024*). The prevalence of depression in the post-conflict population of the Syrian refugees, Nepalese, Mogadishu-Somalia, Ethiopian and Uganda were 59.4%, 27.4%, 59%, 38.3% and 67%, respectively (*Anbesaw et al., 2024*). While a pre-established diagnosis of clinical depression increases the risk of consequent type 2 DM by 60%, a pre-existing diagnosis of Type 2 DM poses a 15% risk of depression in the long run (*De Groot, 2023*). This implicates the potential impact of conflicts in the etiology of depression in the aftermath of conflict.

Ultimately, a bidirectional relationship exists between diabetes and depression (*Berge & Riise, 2015*). For instance, the prevalence of depression among individuals with diabetes mellitus is two to three times higher than that among people without diabetes (*Bădescu et al., 2016*). Chronic stress is a common denominator for both of them (*Bădescu et al., 2016*). Similarly, depression increases the risk of diabetes by 37% (*Knol et al., 2006*). However, this study did not find a statistically significant difference in the prevalences of DM in the two studied communities despite a markedly elevated prevalence of depression of 30% in Mubi

as against 5% in Jada. In addition, the prevalences of DM in both communities were about the same (Mubi 5.44% *vs* Jada 5%). This is consistent with a recent study that reported a prevalence of diabetes ranging from 1.6% to 4.6% in the same region (*Stephen et al., 2024*). By implication, an effective mental health support program in this recovery will in the long run reduce the burden of DM and its attendant complications, which include diabetic kidney disease, heart attack and stroke, which are all NCDs. Furthermore, this study has also shown that DM is also associated with advancing age as reflected in the middle and elderly age brackets in Mubi. This aligns with the findings in a Chinese study (*Bai et al., 2021*) which revealed that the prevalence of type 2 DM increases with age, underscoring the multifactorial origin of NCDs (*Syed et al., 2019*).

In this study, hypertension is associated with increasing age, family history of hypertension, tertiary education, high socioeconomic status, being widowed and diabetes, which further portrays the multifactorial origin of NCDs (*Alsaadon et al., 2022*). Exposure to conflict bears no statistically significant relationship with hypertension in this study. This is contrary to the findings in South Sudan, which showed that the risk of hypertension increased by 50% in its conflict-exposed population (*Narayan, Ali & Koplan, 2010*). This could be due to differences in population characteristics, conflict duration, healthcare access, coping mechanisms, and other study-related differences.

Epidemiologically, hypertension tends to coexist with DM (*Alsaadon et al., 2022*). This coexistence is known to be mediated by several pathophysiologic mechanisms. Chronic inflammation, insulin resistance, oxidative stress, and vascular dysfunction have been implicated as some of the fundamental pathogenesis common to both (*Sharma et al., 2020*). Background family history of diabetes and hypertension is also associated with the eventual diagnosis of these diseases in their first-degree relatives, as reflected in this study (*Moke et al., 2023*). IDPs have increased physical inactivity (*Akinrolie et al., 2022*), and physical inactivity increases the burden of depression.

In this study, a comparatively higher prevalence of abdominal obesity is seen in Jada (59%) when compared to Mubi (31%). Population-based studies have shown that obesity leads to depression and vice-versa (*Steptoe & Frank, 2023*). Conversely, findings from this study did not reflect a corresponding increase in the burden of DM in Jada, which has a higher prevalence of obesity. However, the prevalence of abdominal obesity in both communities surpasses the national obesity prevalence of 15% in Nigeria (*Ramalan et al., 2023*). Obesity is increasingly being implicated as a risk factor for so many diseases, including diabetes (*Ramalan et al., 2023*), hypertension, breast cancer, myocardial infarction and stroke (*Kinlen, Cody & O'Shea, 2017*). Hence, the local primary health centres in Jada should intensify public health interventions such as health education on the need to adopt lifestyle behavioral modifications such as maintaining a healthy weight and exercising.

In this study, there are more people of low socioeconomic status in Mubi than in Jada. Generally, prolonged violent conflicts impoverish communities and reduce both personal income and commonwealth. A study showed that northeastern Nigeria has been severely impacted economically by the insurgency that plagued the region alongside the perennial farmers-herders clashes, which saw many businesses destroyed (*Odozi & Uwaifo Oyelere, 2019*). Low socioeconomic status is a risk factor for both T2DM

and hypertension (*Blok et al., 2022*; *Hill-Briggs et al., 2021*). Consequently, economic devastation occasioned by armed conflict increases the burden of NCDs in the population, hence the need for social safety nets to stem the tide of rising NCDs, by economically empowering the locals through the provision of capital, facilities and skill acquisition support.

### Limitations

The present study was constrained by its hospital-based design, which predominantly involved recruiting individuals from relatively cosmopolitan and suburban locales with access to healthcare facilities. Moreover, we excluded individuals who were frail or bedridden on admission, potentially overlooking cases of NCDs prevalent within this demographic. In addition, this study did not include common cancers which are increasingly constituting some significant proportions of NCDs. Under the Nigerian healthcare structure, the diagnosis and management of cancer are beyond the confines and jurisdiction of general hospitals. Furthermore, cancer diagnosis requires more expertise, funding and technicalities, which are not available in general hospitals where this study was conducted. They basically provide primary care healthcare and some aspects of secondary healthcare services.

Despite these limitations, this study has several strengths, mainly its novelty. To the best of our knowledge, this study is the first conducted to comparatively analyze the impact of conflict on the prevalence of NCDs in Northeastern Nigeria during the post-conflict recovery and rebuilding phases. Moreover, the inclusion of both inpatients and outpatients contributed to the representativeness of the study population, thereby enhancing the generalizability of the study findings to a certain degree.

## CONCLUSION

This study sheds light on the significant disparities in key health indicators between Mubi, a conflict-affected area, and Jada, a more stable environment. The prevalence of depression in Mubi is notably higher than that in Jada, emphasizing the urgent need for mental health support in conflict-affected regions. Limited access to psychosocial support for those with diabetes in Adamawa State further underscores the challenges faced by individuals with NCDs in the region. Lifestyle factors, such as nutrition and exercise, are crucial modifiable risk factors for T2D. This highlights the need for awareness campaigns on healthier dietary practices. Governmental agencies and the private sector are imperative for addressing the rising burden of NCDs in Nigeria and worldwide. This research underscores the urgent need for targeted interventions, policy changes, and increased resources to combat the growing challenge of NCDs, especially depression, in conflict-affected regions.

### Funding

The authors received no funding for this work.

## Competing Interests

Oyelola Adegboye is an Academic Editor for PeerJ.

## Author Contributions

- Roland I. Stephen conceived and designed the experiments, performed the experiments, authored or reviewed drafts of the article, and approved the final draft.
- Jennifer A. Tyndall conceived and designed the experiments, performed the experiments, authored or reviewed drafts of the article, and approved the final draft.
- Jamiu S. Olumoh performed the experiments, analyzed the data, prepared figures and/or tables, authored or reviewed drafts of the article, and approved the final draft.
- Malachy I. Okeke performed the experiments, authored or reviewed drafts of the article, and approved the final draft.
- Jacob A. Dunga performed the experiments, authored or reviewed drafts of the article, and approved the final draft.
- Tonde G. Elijah performed the experiments, authored or reviewed drafts of the article, and approved the final draft.
- Dillys M. Bello performed the experiments, authored or reviewed drafts of the article, and approved the final draft.
- Oyelola A. Adegboye performed the experiments, analyzed the data, prepared figures and/or tables, authored or reviewed drafts of the article, and approved the final draft.
- Jimmy A. Reyes performed the experiments, authored or reviewed drafts of the article, and approved the final draft.

## Human Ethics

The following information was supplied relating to ethical approvals (i.e., approving body and any reference numbers):

National Health Research Ethics Committee, Federal Ministry of Health, Abuja, Nigeria

## Data Availability

The raw measurements are available in the Supplementary File.

## Supplemental Information

Supplemental information for this article can be found online at http://dx.doi.org/10.7717/peerj.18520#supplemental-information.

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
