# Peer review of "The pattern and burden of non-communicable diseases in armed conflict-exposed populations in Northeastern Nigeria"

_PeerJ, doi:10.7717/peerj.18520_

## Round 0.1 · original submission · Major Revisions

The subject of the article is very interesting. However, some gaps reported by the reviewers have substantially affected the quality of the manuscript.

I also have other comments related mostly to the introduction and the discussion.
The introduction ideas should be reorganized, and you should also provide the link between the prevalence of NCDs and the conflict in the region.

The same for the discussion: you should discuss your results, not just provide the results

Reviewer 1 ·

Basic reporting

The manuscript studies an interesting topic about the NCDs in armed conflict-exposed populations in Northeastern Nigeria. This manuscript follows a conventional structure, which facilitates the comprehension of the study's aims, methods, results, and conclusions. Below are some detailed comments:

Line 49-54: please report the aOR, their CIs, IRR, and their Cis using the same digit place for consistency reporting. Please check all the other numbers throughout the manuscript and the Tables, figures if applicable.

Line 68-70: The term "several residents" is vague. Please provide an estimate for more precision.

Line 89-96: While you have discussed the importance of studying the prevalence of depression in your introduction, the motivation and current status of the other three NCDs—hypertension, diabetes, and obesity—have not been adequately addressed. Please provide a more detailed introduction for all four NCDs examined in this study, including their prevalence, impact, and relevance in the context of conflict-exposed populations in Northeastern Nigeria. This will help to establish a comprehensive background and justification for your research.

Line 107: Please be specific about the “region”.

Line 124-126: Please be specific about the four months period. When did this investigation start and when did it end?

Line 130-131: Will the trained data collectors help the participants if the participants were illiterate?

Line 143 Please provide a citation for this.

Line 158 Please specify what size is a number of servings for fruit and vegetable intake.

Line 166-168, Line 169-170 Please cite those two website using appropriate citation format

Line 206-207: please specify how the model accounted for the complex survey design. Did you use stratification and clustering in your models?

Line 217: Please report the p values using three digits throughout the manuscript and including the tables.

Table 1.
1. Please report age using the categorization you used in the model as well. Also I noticed that in the questionnaire of supplementary material, you collected the categorical age instead of continuous age.
2. Please report all the p values in 3 digits, please report p values that are less than 0.001 into “<0.001”. Please also update the manuscript as well such as line 228 and line 232
3. Please clarify how the low, middle, and upper income in the socioeconomic status
4. Please report *Calculated using…. Significant.” In the end of the table 1 as footnote

Line 271 Please discuss how you categorized your age.

Line 271-283 Please add reference group when you interpret your model results

Experimental design

The experimental design is generally well-structured. Here are some detailed comments:

Line 126-127: You mentioned that you used random sample. Please be specific about how you conduct your random sample process, please note that random sampling is not the same with convenience sampling.

Line 132-137: You have used the Fischer’s formula for the sample size calculation for hypertension.
1. You should make sure you have the enough sample size for other NCD outcomes as well
2. You did not mention what the d stands for in the formula

Lines 177-182: The procedure description is adequate, but more detail on the data collection methods, including any tools or instruments used, would enhance reproducibility. Mention the training process for data collectors to ensure data quality.

Validity of the findings

The methods employed are suitable for the study design, and the results have been reported clearly. However, there are several points should be addressed to insure the validity of the finding. Here are some detailed comments:

Line 194-197, Line 251-Line256: Please specify how the matching was conducted and indicate the proportion of data that remained unmatched. Additionally, it would be beneficial to present the detailed results after propensity score matching. This information will provide a clearer understanding of the robustness and validity of your findings.

Line 204: The term "incidence rate" typically refers to events per person-time, and "incidence rate ratio" is the ratio of two incidence rates from different groups. Please use appropriate terminology for the effects derived from the complementary log regression model. Additionally, clarify the acronym "MClog-log" by specifying what the "M" stands for and ensure you are using the correct model. It is also recommended to report logistic regression models for the depression and diabetes analyses as a sensitivity analysis and include these model results in the supplementary material.

Line 354-359: You sampled your participants from two hospitals, which poses a significant limitation. Your findings may not be generalizable to the broader population, as the data were collected exclusively from hospital patients rather than the general population.

Table 3
I looked into your data, you have not considered the missing in the data. For example, I noticed that for the question “Which of these chronic disease(s) conditions do you have?”, there were 45 patients reported “I don’t know”. Those 45 patients should not be coded as No for the NCDs, therefore, your multivariable models should at most have 973-45=928 patients. This did not consider the missing values in your other variables, so your model could have less patients than 928 patients. However, your model reported 970 observations for the obesity and hypertension models. And also please report the number of observations for other two models as well.
A p-value <0.05 from the goodness-of-fit test suggests poor model fitting, please discuss how this impacts the validity of your finding.

Line 310 Please use the word multivariable for consistency

Line 313 Please use the same acronym RR vs IRR

Reviewer 2 ·

Basic reporting

Stephen et al presented a cross-sectional study to investigate the prevalence and burden of non-communicable diseases in armed conflict-exposed populations in Northeastern Nigeria, compared with non-conflict regions. And I have several other comments listed below for the authors to consider.

Experimental design

1) The authors mentioned that this was an unmatched study, but then applied propensity score matching in section 2.5. Further, the results in line 254 “… with respective p-values of 0.438, 0.849, and 0.070.” were not listed in any tables or appendix.

Please align and revise these accordingly.

2) In line 261-270, the authors mentioned “The participants from the conflict-exposed Mubi exhibited significantly lower odds of abdominal obesity than their counterparts in Jada (control)”, followed by “Conversely, the participants in the conflict-exposed Mubi showed significantly higher odds of having abdominal obesity.” These two statements are self-contradictory.

And from table 3, “Affected by insurgency” does not represent the Mubi region?

Please examine the corresponding results and revise expressions for clarity.

Validity of the findings

1) Any particular rationale for using OR for obesity and hypertension, while using RR for depression and diabetes?

2) Since cancer is also one component of NCDs, the author should also address the limitation and rationale of not including cancer into the outcomes of interest in this study.

Additional comments

1) Please provide the related references in line 143, regarding the World Health Organization definition on hypertension cutoff.

2) Line 144 – please add the unit “mmHg” for hypertension measurement.

·

Basic reporting

Generally, the manuscript is well-written and uses clear, appropriate, and scientific language. The structure and reporting follow acceptable professional standards. The background literature review contains relevant and recent evidence. However, some assertions are not supported by relevant literature, and some points discussed are irrelevant to the study's objective. For example, the three sentences beginning the first paragraph of the Introduction, starting with "The northeastern part of Nigeria, which was plagued by insurgency over the last 66 14 years…" made several uncorroborated statements. Similarly, the assertion, "Living in humanitarian situations for approximately a decade took a toll on the displaced persons and caused them serious psychological distress", in Lines 70-71, is not supported with evidence. Moreover, I would like to see empirical evidence from Nigeria to support the burden of NCDs in Nigeria.
Some of the in-text citations were not correctly cited; for example, Jawad et al. (Line 89), Badescu et al. (Line 300 ), and Li et al. (Line 327). In Line 89, the authors cited Jawad et al. (2019) in the intext and cited another study in the Reference. The same mistake was made with the Reference that follows this, where a study from Ukraine was cited. Please check that all your in-text citations are correctly matched with their Reference.
I could not confirm if all appropriate raw data have been made available in accordance with the journal's data-sharing policy.

Experimental design

The aim/objective of the study is clearly stated, preceded by the gap (contextual lack of research) in the literature and how the study will fill the gap. It was then followed by the practical implication of the study.
The study used descriptive and regression methods to answer the study's objective. An appropriate technique was used to determine the sample size. I was surprised that the author decided to use the prevalence of hypertension in Ukraine while there are several studies conducted on the prevalence of hypertension in Nigeria, in general, and in the Northern part of the country, in particular. There is even a study conducted in an area close to where this study was conducted [1] Ethical approval was obtained from the relevant agent for the conduct of the study. All the variables chosen for the study are appropriately selected and measured, and the regression methods used are appropriate and relevant.
The authors claimed, "All models accounted for the complex survey design, including stratification and clustering, as well as individual-level factors", without explaining how this was done.
Overall, the methodological approaches chosen are well-described and have sufficient information to be reproducible.

References
1. Olivier PM, Adonis KK, Joachim C. Hypertension and classical risk factors in ambulatory patients: A hospital-based study in Adamawa region (Northern Cameroon). International Journal of Medicine and Medical Sciences. 2011 Oct 31;3(11):331-6.

Validity of the findings

The results are well-presented using tables that were appropriately labelled. The statistical parameters of the results are reported and explained according to standard. Only results relevant to the study's objective were presented in a statistically sound manner.
The Discussion section of the article needs a complete overhauling. The aim of this section is to present a summary, interpretation, implication, limitation, and recommendation based on the findings of a study. Therefore, it must not repeat the result such as done, for example, in Lines 310-31: "Further multi-variant logistic regression analysis conducted in the present study indicated that the relative risk for diabetes among the participants from Mubi was 2.13 (95% CI: 0.882ñ5.132)". Also, results must be compared and contrasted with relevant studies. This is obviously lacking in the Discussion of this study. For example, many assertions were made without supporting literature evidence (for example, Sentences in Lines 319-321 and the opening sentences of the paragraphs in Lines 333-340 and Lines 341-343), and many studies referenced are not related to the findings of the study (for example, the research on malnutrition in the US (Lines 336-340). Moreover, unexpected findings are expected to be highlighted in the interpretation of the findings and the possible reason for the deviation suggested.
To my surprise, it appears the study was so perfectly conducted that it has just only two limitations! However, I am impressed by the recognition in Lines 346-347 that the pandemic has shifted focus away from NCD research. Nevertheless, you might do better by supporting this with relevant evidence.
The conclusions drawn from the findings are appropriate and relevant. However, the statement in Lines 376-379 should be expunged from the Conclusion and moved to the main Discussion since this was not mentioned in the Discussion. The Conclusion should not be an orphan. It must sprout from the findings discussed in the main body of the Discussion.

Additional comments

4. General comments
This study would be suitable for publication if the revisions suggested are carried out. Overall, the study is novel, relevant and conducted using standard methodology, and the results are validly presented.

---

## Round 0.2 · Minor Revisions

The authors made considerable eforts to improve the quality of the manuscript. However, some concern still be present:

1. The introduction needs (again) to be reorganised.
2. the manuscipt needs to be edited for some spelling errors (see the atached file).

Reviewer 1 ·

Basic reporting

The authors have made significant strides in addressing the concerns raised in the previous review.

Experimental design

The authors have made significant strides in addressing the concerns raised in the previous review.

Validity of the findings

The authors have made significant strides in addressing the concerns raised in the previous review.

·

Basic reporting

There has been a significant improvement in the manuscript over the previous version. The Introduction has been improved with relevant and correct citations. However, a couple of spacing and spelling errors were observed. I have noted this in the tracked manuscript file. In addition, I have added some important comments in the discussion section.

Experimental design

The methodology used in the study has been explained and improved with more detailed clarity.

Validity of the findings

The Result and Discussion (including Conclusion) sections have been significantly improved. A few areas requiring improvements have been noted in the tracked manuscript file.

Additional comments

With the attention to the highlighted corrections to be made in the manuscript, I have no hesitation that the manuscript should be accepted for publication.

---

## Round 0.3 · accepted · Accept

The authors have responded to all the editor and reviewers comments